# A Blend of F Prostaglandins Functions as an Attractive Sex Pheromone in Silver Carp

**Peter W. Sorensen** [1,*], **Mara C. P. Rue** [1,2], **Joseph M. Leese** [1,3], **Ratna Ghosal** [1,4] **and Hangkyo Lim** [1,5]

1. Department of Fisheries, Wildlife and Conservation Biology, University of Minnesota, 2002 Upper Buford Circle, St. Paul, MN 55108, USA; mcprue@brandeis.edu (M.C.P.R.); Joseph.Leese@desales.edu (J.M.L.); Ratna.Ghosal@ahduni.edu.in (R.G.); hlim@ndm.edu (H.L.)
2. Volen Center and Biology Department, Brandeis University, Waltham, MA 02453, USA
3. Biology Department, DeSales University, Center Valley, PA 18034, USA
4. Biological and Life Sciences, School of Arts and Science, Ahmedabad University, Ahmedabad, Gujarat 380009, India
5. Department of Biology, Notre Dame of Maryland University, Baltimore, MD 21210, USA
* Correspondence: soren003@umn.edu; Tel.: +1-612-624-4997

**Abstract:** A series of laboratory experiments tested the hypothesis that the Silver Carp (*Hypophthalmichthys molitrix*), an invasive river carp from China, employs a prostaglandin $F_{2\alpha}$-derived sex pheromone that is attractive and species-specific. Using electro-olfactogram recording (EOG), we found that the olfactory system of this species is acutely sensitive to three F-series prostaglandins (PGFs) at picomolar concentrations and that this sensitivity is enhanced when juveniles are masculinized using androgens, consistent with expectations of a sex pheromone. Individual PGFs had behavioral activity but it was low, suggesting a mixture might be important. To pursue this possibility, we implanted carps with osmotic pumps containing prostaglandin $F_{2\alpha}$ ($PGF_{2\alpha}$), a treatment previously shown to elicit release of a PGF-based spawning pheromone in the Common Carp. We found that $PGF_{2\alpha}$-implanted Silver Carp released a species-specific odor that contained a blend of $PGF_{2\alpha}$ and two of its metabolites, which masculinized individuals detected and were attracted to with high sensitivity. Finally, we found that a mixture of these PGFs was attractive to masculinized Silver Carp, while a different mixture released by Bighead Carp was not. We conclude that Silver Carp likely use a species-specific $PGF_{2\alpha}$-derived sex pheromone that is probably released at spawning and might be useful in its control. Confirmatory studies that explore pheromone function in naturally mature Silver Carp using natural odors in the field should now be conducted to further confirm our proof-of-concept study.

**Keywords:** pheromone; invasive species; spawning; attractant

## 1. Introduction

Pheromones are chemical cues released by animals and many other organisms that drive behavioral and/or physiological changes in conspecifics in adaptive manners [1–3]. Living in aquatic environments that typically offer few visual cues, teleost fishes have commonly evolved to employ pheromones to mediate many aspects of their life histories [4]. Sex pheromones are particularly important in the regulation of reproduction and can serve both priming (physiological) and/or releasing (behavioral) roles [5,6]. With only one possible exception [7], all teleost sex pheromones described to date have been found to contain sex hormones and their derivatives [5,6]. Hormonal sex pheromones have now been identified in about half a dozen model fish species including two Eurasian Carps (Goldfish, *Carassius auratus*, and Common Carp, *Cyprinus carpio*), a few Cichlids including the

tilapias, *Oreochromis sp.*, some gobies, and salmonids [6,8]. Being potent attractants, species-specific, and easily added to the water, sex pheromones are candidates for possible application as attractants for invasive fishes to facilitate their removal or quantification [9].

Sex pheromones in teleost fishes are perhaps best understood in two Eurasian Carps, the Goldfish and its relative the Common Carp [6,10,11]. Females of these seasonal scramble spawning Cyprinid carps [10] employ a mixture of several sex steroid hormones and their metabolites as a priming pheromone to synchronize egg and sperm maturity during the perio-ovulatory period, and then a mixture of prostaglandin $F_{2\alpha}$ ($PGF_{2\alpha}$) and its metabolites, 15keto-prostaglandin $F_{2\alpha}$ ($15K\text{-}PGF_{2\alpha}$) and 13,14dihydro-15keto-$PGF_{2\alpha}$ ($dh15K\text{-}PGF_{2\alpha}$) as a spawning (releasing) pheromone [6,10,12]. $PGF_{2\alpha}$ is known to be produced by ovulated Goldfish as a hormone to stimulate female sexual behavior before functioning as a sex pheromone that is released within minutes of ovulation and spawning [12–14]. The PGF pheromone both attracts males and signals that females are sexually receptive and carrying eggs, stimulating courtship. Both field and laboratory studies using Goldfish and Common Carp have shown that PGF-pheromone release can also be induced in immature fish by injecting them with $PGF_{2\alpha}$ or implanting them with osmotic pumps containing $PGF_{2\alpha}$, which is then metabolized and released along with other natural products including two metabolites, $15K\text{-}PGF_{2\alpha}$ and $dh15K\text{-}PGF_{2\alpha}$ [12,13]. Notably, $PGF_{2\alpha}$-implants mimic natural circulating levels of $PGF_{2\alpha}$, eliciting normal pheromone release for days and making this technique very useful as well as biologically relevant [12]. Electrophysiological experiments have also shown that the olfactory systems of both the Goldfish and Common Carp and a variety of other fishes detect $PGF_{2\alpha}$ and its metabolites, $15K\text{-}PGF_{2\alpha}$ and $dh15K\text{-}PGF_{2\alpha}$, at low concentrations [13,15] in a sexually dimorphic manner that is inducible by androgen treatment [16–18]. Studies of the PGF spawning pheromone show that Goldfish and Common Carp discern each other's odors and that a variety of PGFs as well as other metabolites are responsible while specific ratios of PGFs may not be especially important [11]. Finally, the activity of the PGF sex pheromone has been confirmed in the field using $PGF_{2\alpha}$-implanted Common Carp odor [12], suggesting this technique might be used as a tool to facilitate attracting, monitoring, and trapping these invasive fishes [9].

Two congeneric species of carp from Asia, the Silver Carp (*Hypophthalmichthys molitrix*) and the Bighead Carp (*H. nobilis*), together known as Bigheaded Carps, and globally some of the most commonly cultured species [19], were introduced in the United States in the late 1970s, escaped, and have become highly invasive in the Mississippi River [20]. Although both of these Asian Carp species share many similarities with the Common Carp, they represent a distinctly different evolutionary branch that diverged several million years ago [21,22]. These naturally sympatric Asian Carps are now found in great abundance throughout most of the Mississippi River, and techniques to monitor their distribution and control them are sought. Both Bigheaded Carp species typically live in productive, turbid river waters and appear to have very well developed olfactory and internal gustatory systems [23,24] and reproduce episodically in a highly synchronized fashion in flowing turbid waters [25,26], suggestive of a pheromonally-mediated reproductive strategy. This suggests that they, like the Eurasian carps, employ hormonal sex pheromones, and PGFs in particular, to promote reproductive synchrony and that this trait could be exploited in their control. However, the possibility that Asian carps use F-series prostaglandins (PGFs) or other hormonal products as pheromones has not yet been examined.

The present study asked the question of whether the Silver Carp might also use PGFs as a sex pheromone to attract and discern conspecifics. Because this species naturally matures at a large size (about 60 cm) and is very difficult to study in the field, we conducted a proof-of-concept laboratory study with small (juvenile) fish using the same techniques that we have developed to artificially masculinize and study reproductive behaviors in the Goldfish as well as some other cyprinids [17,18]. We asked five questions. First, is the Silver Carp olfactory system acutely and specifically sensitive to PGFs and can it be increased by masculinization with androgens (as expected of a sex pheromone)? Second, are these individual PGFs highly attractive to masculinized fish? Third, when implanted with $PGF_{2\alpha}$, do Silver Carp release a mixture of $PGF_{2\alpha}$ and its metabolites, and is this mixture profile

distinctive, as might be expected if these PGFs function together as a mixture (blend)? Fourth, is the odor of $PGF_{2\alpha}$-implanted Silver Carp and blend of PGFs they release attractive to male (masculinized) conspecifics and in a species-specific manner? Fifth, do the mixture profiles of PGFs released by the $PGF_{2\alpha}$-implanted carps explain the behavioral specificity their pheromonal odor might have?

## 2. Results

Electro-olfactogram (EOG) recording from the olfactory systems of Silver Carp established that they are highly and specifically sensitive to PGFs and that this sensitivity is influenced by androgens and thus likely sexually-dimorphic. Initial studies of the olfactory sensitivity of untreated juvenile Silver Carp established that while these fish are insensitive to E series prostaglandins (Supplementary Data, Supplemental Figure S1), they detected all three F series prostaglandins previously identified as pheromones in the Goldfish: $PGF_{2\alpha}$, $15K-PGF_{2\alpha}$, and $dh15K-PGF_{2\alpha}$ with high sensitivity. Further, while there was no apparent difference between the olfactory sensitivities of untreated juvenile fish and ethanol-treated control fishes to PGFs (their 95% confidence intervals overlapped), the olfactory sensitivity (EOG responses) of masculinized Silver Carp treated with methyltestosterone (MT) were greater than controls to $10^{-8}$ Molar (M) $PGF_{2\alpha}$ ($p = 0.05$; df = 2), $10^{-8}$ M $15K-PGF_{2\alpha}$ ($p = 0.02$; df = 2), and $10^{-8}$ M $dh15K-PGF_{2\alpha}$ ($p = 0.03$, df = 2) (Figure 1). Specifically, MT-treated fish detected $PGF_{2\alpha}$ down to a concentration of approximately $10^{-11}$ M (i.e., their confidence intervals did not overlap the y axis) with responses rising to about twice that of the standard, $10^{-5}$ M L-Serine (a food odor) at $10^{-8}$ M, nearly twice the size response noted in the control carp to this concentration. Responses to $15K-PGF_{2\alpha}$ were even larger; MT-treated carp detected this PGF down to approximately $10^{-12}$ M and showed a response nearly three times that of the standard at $10^{-8}$ M. Responses to $dh15K-PGF_{2\alpha}$ were similar to those elicited to $15K-PGF_{2\alpha}$: MT-treated carp detected it down to approximately $10^{-12}$–$10^{-11}$ M and also responded to $10^{-8}$ M with an EOG about three times the size of the standard. EOG tests of whether binary mixtures of PGFs are detected by more than one type of olfactory receptor suggested that $PGF_{2\alpha}$ is detected by its own independent olfactory receptor (the mean mixture discrimination index [MDI] for a mixture of $PGF_{2\alpha}$ and $15K-PGF_{2\alpha}$ was $1.39 \pm 0.7$—different from 1.0 ($p = 0.02$, df = 2), the expected MDI if they shared a receptor—and the MDI for $PGF_{2\alpha}$ and $13dh15K-PGF_{2\alpha}$ was $1.34 \pm 0.12$ vs. 1.0 ($p = 0.05$, df = 2). Conversely, it was not clear whether $15K-PGF_{2\alpha}$ and $13dh15K-PGF_{2\alpha}$ have their own receptors (the MDI for a mixture of $PGF_{2\alpha}$ and $dh15K-PGF_{2\alpha}$ was $1.12 \pm 0.12$ vs. 1.0; $p > 0.05$, df = 2). When EOG responses to the PGFs were measured after a year of MT treatment and at the conclusion of behavioral experiments (see below), they were found to be about twice the size they were after 3 months of MT treatment (Supplementary Data, Supplemental Figure S2).

Tests of the behavioral activity individual PGFs in a circular maze showed that 10 nanomolar (nm) concentrations of each of the three PGFs attracted MT-treated Silver Carp ($p = 0.02$, $p = 0.1$, $p = 0.01$, df = 9) when tested on their own and that control (EtOH-treated) Silver Carp were not attracted to $15K-PGF_{2\alpha}$ ($p > 0.05$; df = 10), further suggesting these compounds function as a sex pheromone(s) (Figure 2). Nevertheless, these responses were very small (only about 10% of the fish were measurably attracted), suggesting that on their own, these individual PGFs might not be the entire pheromonal signal.

Further investigation of $PGF_{2\alpha}$ metabolism and release in Bigheaded carps to determine if mixtures of PGFs might instead function as the sex pheromone showed that they, like the Goldfish and Common Carp, release substantial but different quantities of all three PGFs ($PGF_{2\alpha}$, $15K-PGF_{2\alpha}$, and $dh15K-PGF_{2\alpha}$) to the water when implanted with $PGF_{2\alpha}$ (Table 1). Notably, Silver Carp released approximately a third less $PGF_{2\alpha}$ than either the Bighead or Common Carp. Nevertheless, $PGF_{2\alpha}$ dominated all three mixtures with Silver Carp water containing more $15K-PGF_{2\alpha}$ and $dh15K-PGF_{2\alpha}$ than the other two species' release waters. Chi-square analysis showed that the quantities of the three PGFs varied between species ($X^2 = 35.2$; $p = 0.01$), but that the relative amounts of the PGFs released by Common Carp and Bighead Carp did not differ from each other ($X^2 = 0.426$; $p > 0.05$). When

corrected for mass (molarity), the Silver Carp PGF release ratio (PGF$_{2\alpha}$:15K-PGF$_{2\alpha}$:dh15K-PGF$_{2\alpha}$) was 94.7:2.3:2.7; Bighead Carp was 99.0:0.1:0.8; and Common Carp was 99.4:0.1:0.5 (after rounding).

When the behavioral activity of PGF$_{2\alpha}$-implanted carp odor was next tested in a rectangular maze designed to facilitate natural shoaling behavior using masculinized Silver Carp, we found that fish were strongly attracted (spending 97% of their time) to the holding water (odor) of PGF$_{2\alpha}$-implanted conspecifics ($p = 0.03$, df = 6) but ignored control well water (Figure 3A). In contrast, Silver Carp did not appear to respond to either the odor of PGF$_{2\alpha}$-implanted Bighead Carp odor, or the odor of PGF$_{2\alpha}$-implanted Common Carp water (or well water controls) ($p > 0.05$; df = 6; Figure 3B,C).

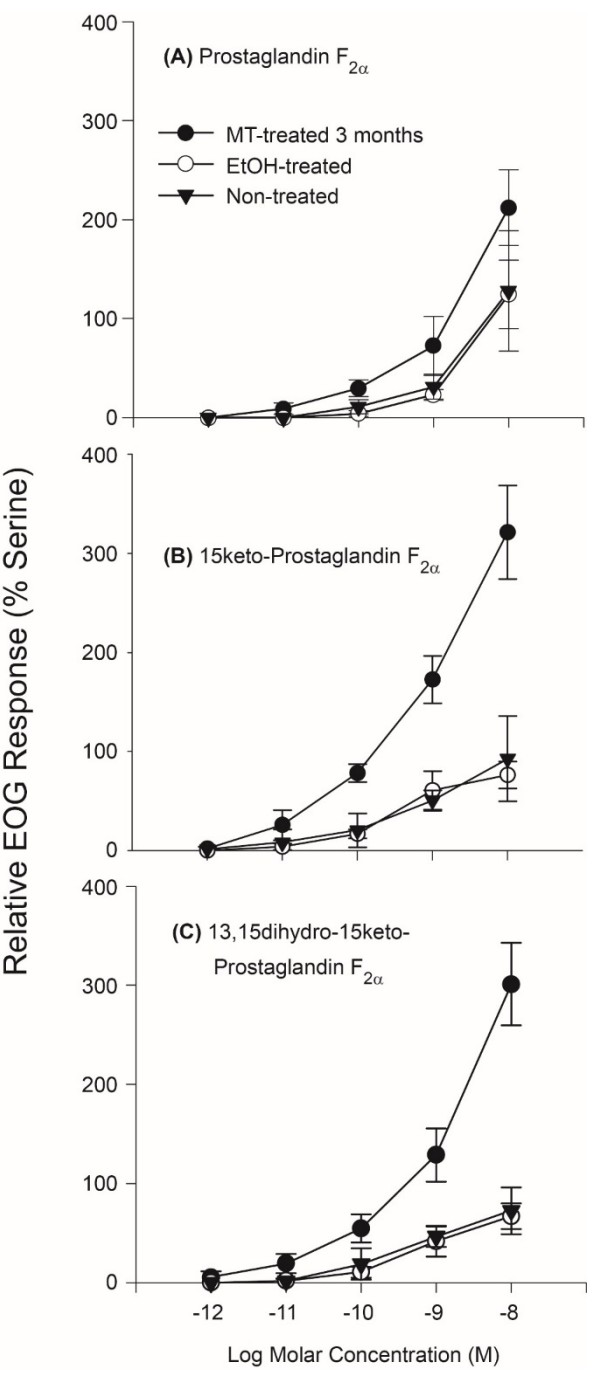

**Figure 1.** Mean olfactory responses (±95% confidence intervals) of untreated, control, and MT (masculinized) Silver Carp to: (**A**) PGF$_{2\alpha}$, (**B**) 15K-PGF$_{2\alpha}$, and (**C**) dh15K-PGF$_{2\alpha}$, as measured by EOG recording (n = 3 for each group). Response sizes are expressed relative to the L-Serine standard.

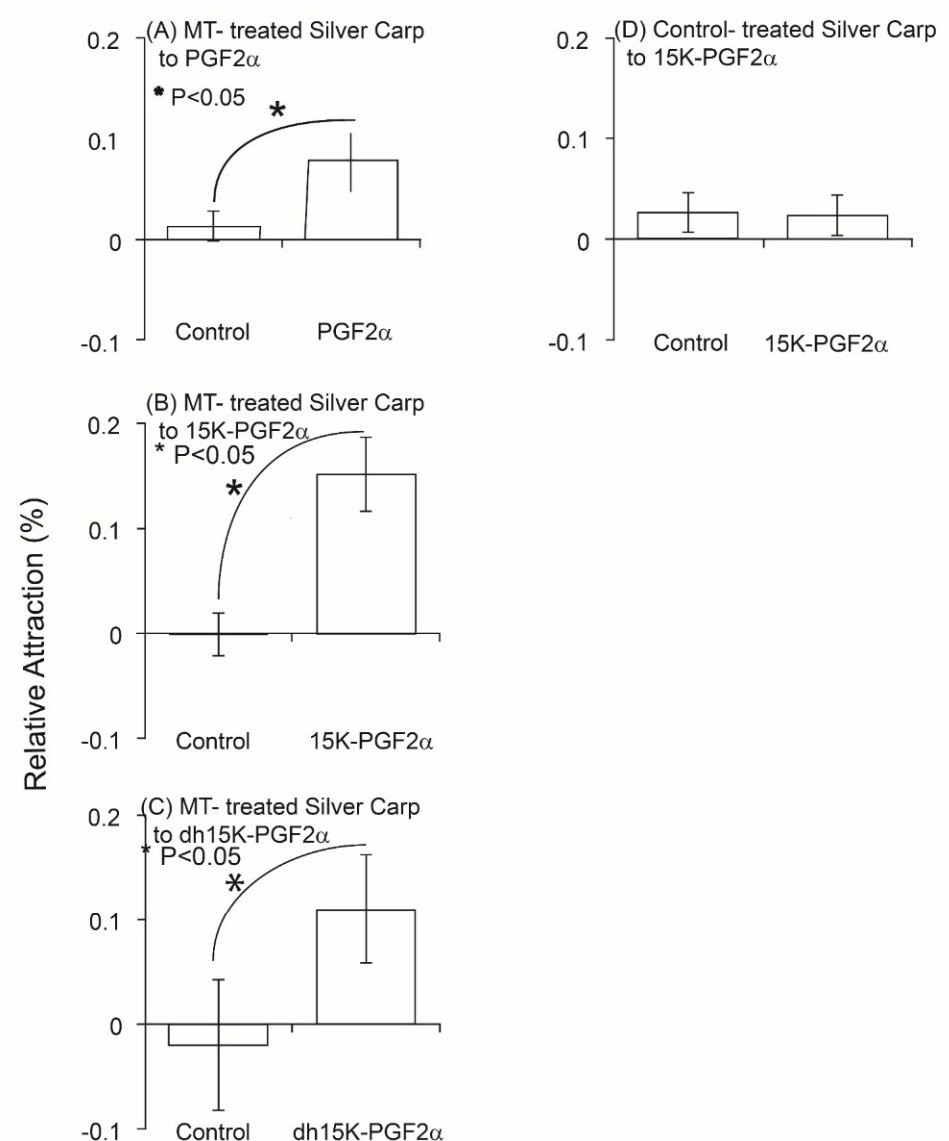

**Figure 2.** Relative distribution (%) of masculinized (MT-treated) and non-masculinized Silver Carp in a circular test maze when exposed to water alone (control) or to each one of three individual PGFs (10 nanomolar concentrations). Each experiment had a control 15-min test period during which time well water alone was added followed by a 15-min period when a PGF was added. Differences in time spent between trials were calculated and shown. n = 12 trails for each. The three test stimuli were: (**A**) $PGF_{2\alpha}$, (**B**) $15K\text{-}PGF_{2\alpha}$, and (**C**) $dh15K\text{-}PGF_{2\alpha}$. Control (**D**) (non-masculinized Silver Carp) were also tested with $15K\text{-}PGF_{2\alpha}$ and were not attracted ($p > 0.05$). * $p < 0.05$.

**Table 1.** Mean release rates ng/g fish/h ($\pm$S.E.) of the three PGFs from 50 g ($\pm$10 g) Silver, Bighead, and Common Carps implanted with $PGF_{2\alpha}$-filled osmotic pumps. n = 3.

|  | PGF Type | | |
| --- | --- | --- | --- |
| Species | $PGF_{2\alpha}$ | $15K\text{-}PGF_{2\alpha}$ | $dh15K\text{-}PGF_{2\alpha}$ |
| Silver Carp | 425.3 (11.0) | 10.7 (0.85) | 11.9 (1.71) |
| Bighead Carp | 630.6 (33.4) | 0.6 (0.29) | 5.4 (0.45) |
| Common Carp | 607.4 (48.6) | 0.9 (0.48) | 2.6 (0.22) |

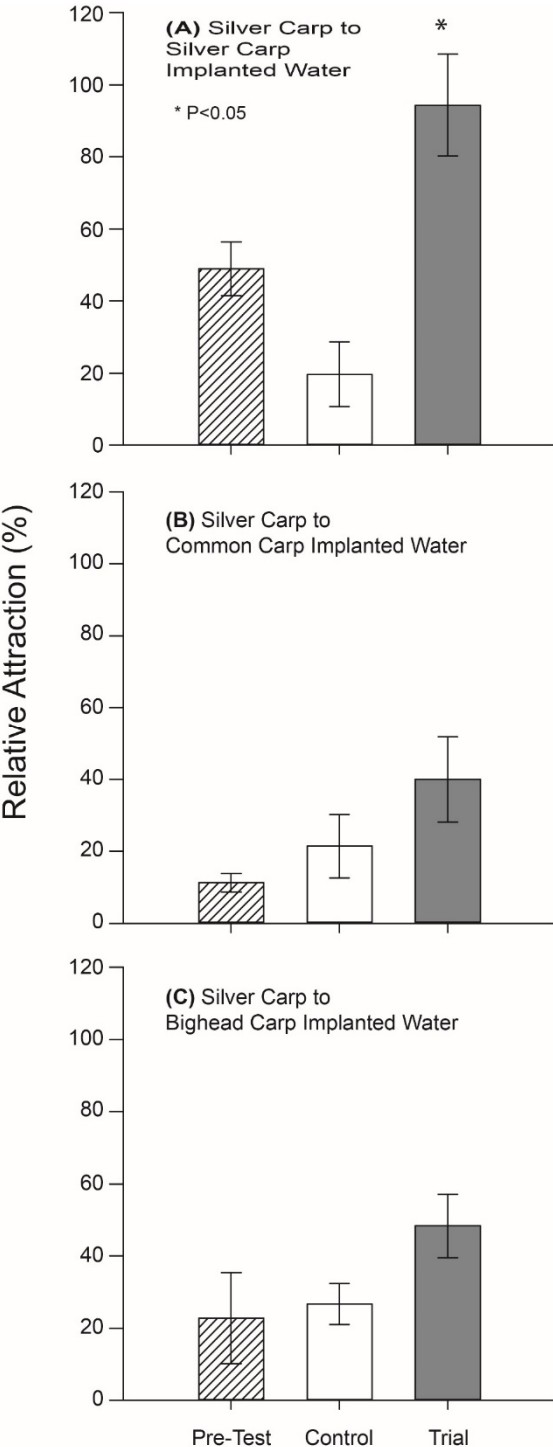

**Figure 3.** Distribution of masculinized Silver Carp in the rectangular test tank before and during tests of $PGF_{2\alpha}$-implanted carp odors. Mean percent time ($\pm$standard error) on the side to which the test odor was added is shown. Each experiment had three 10-min test periods during which time no stimulus was added (striped bar), following one in which well water control was added (open bar), followed by one in which a test stimulus was added (solid bar). Three sets of test stimuli were tested on different fish as follows: (**A**) $PGF_{2\alpha}$-implanted Silver Carp holding water, (**B**) $PGF_{2\alpha}$-implanted Bighead Carp holding water, and (**C**) $PGF_{2\alpha}$-implanted Common Carp holding water. n = 7 for each test. * $p < 0.05$ by 1-tailed paired *t*-test.

Finally, we confirmed that much of the behavioral activity of $PGF_{2\alpha}$-implanted Silver Carp was associated with the blend of PGFs it contained; masculinized Silver Carp were strongly attracted to the mixture of PGFs found in the holding water of $PGF_{2\alpha}$-implanted conspecifics ($p = 0.01$, df = 6), but not the mixture of PGFs found in the holding water of implanted Bighead Carp (Figure 4). Well water again had no apparent effect ($p > 0.05$, df = 6).

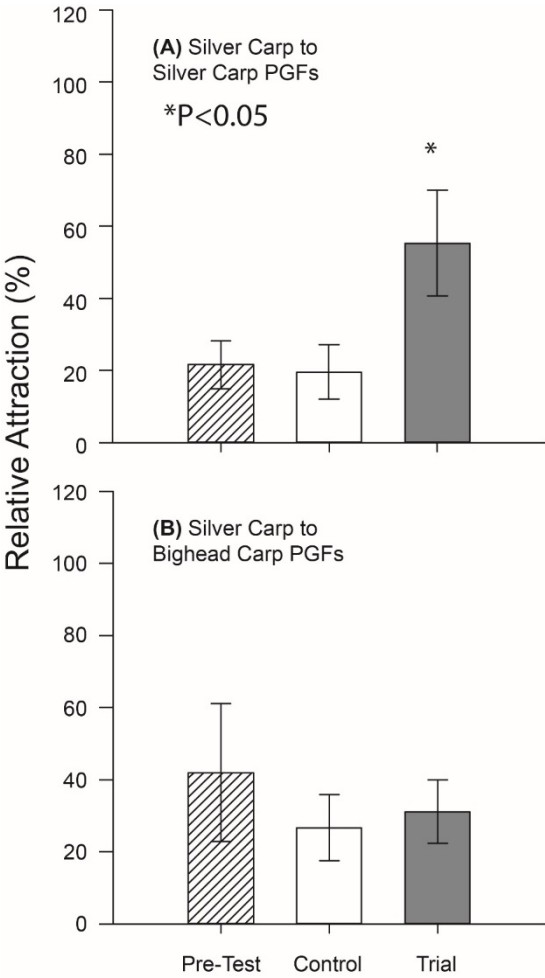

**Figure 4.** Distribution of masculinized Silver Carp in the rectangular test tank during tests of the PGFs found in $PGF_{2\alpha}$-implanted carp odors. Mean percent time ($\pm$standard error) on the side to which the test odor was added is shown. Each experiment had three 10-min test periods during which time no stimulus was added (striped bar), following one in which well water control was added (open bar), followed by one in which a test stimulus was added (solid bar). Two test stimuli were tested as follows: (**A**) the PGFs mixture profile found in $PGF_{2\alpha}$-implanted Silver Carp holding water, and (**B**) the PGFs mixture profile found in $PGF_{2\alpha}$-implanted Bighead Carp holding water. n = 7 for each test. * $p < 0.05$ by 1-tailed paired *t*-test.

## 3. Discussion

Using several well-established approaches, our proof-of concept laboratory study demonstrated that similar to the Eurasian carps, the Silver Carp likely uses a mixture of F prostaglandins as a sex pheromone which almost certainly is associated with spawning. In particular, we found that the olfactory systems of juvenile Silver Carp are remarkably sensitive to three F prostaglandins and that both this physiological and behavioral sensitivity is increased by androgen treatment. Further, at least two of the three PGFs are discriminated from each other. Additionally, masculinized fish were attracted to a mixture of the three PGFs, which we also showed to be released by conspecifics treated with $PGF_{2\alpha}$

following established techniques known to mimic natural pheromone release in Eurasian Carps [11,12]. Very likely, this sex pheromone is normally released in the urine by ovulated, sexually receptive Silver Carp females, as is the case for the Goldfish and Common Carp [27], for which $PGF_{2\alpha}$ is also intimately linked to the process of ovulation and spawning behavior [5,14]. As with the Common Carp, the Silver Carp pheromone signal appears to be specific, although in the case of the Silver Carp the ratio (blend) of PGFs seems to be important. Nevertheless, because ours is a relatively small proof-of-concept laboratory study, this sex pheromone's precise identity and function must now be confirmed in wild, naturally-ovulated free-ranging fish. This sex pheromone could be a very useful tool in Bigheaded Carp management and control as these fish are usually very difficult to locate and track in large flowing rivers [9].

The Silver Carp joins approximately a dozen other fish species for which there is now evidence that PGF-based sex pheromones mediate reproductive behaviors, opening new doors for managing these invasive carp. It, however, is only one of a handful of species for which olfactory sensitivity, behavioral responsiveness, and PGF release have all been demonstrated and is the first Asian Carp species to be so described. Several apparent similarities and differences with Eurasian Carps are evident in this proof-of-concept study. First, while the overall olfactory sensitivities of the Bighead and Eurasian Carps to the three PGFs are similar, differences exist in the way they detect each of them. In particular, the concentration-response relationship to $PGF_{2\alpha}$ appears different between the Eurasian and the Asian (Bigheaded) species because, while it does not saturate in the former species, it does appear to in the latter species. Differences are also seen in responses to $15K-PGF_{2\alpha}$ and $dh15K-PGF_{2\alpha}$. Silver Carp respond to both of these PGFs in a very similar manner and possess a shared olfactory receptor(s) for them, while the Goldfish (the best studied example), responded to these PGFs in different ways. These observations both suggest that these species likely use PGF mixtures in slightly different ways and that the Silver Carp olfactory neurons may use different secondary messenger systems [28]. This would be interesting and important to determine in the future with more definitive studies of receptor and species-specificity that test natural odors in naturally mature fishes. Pilot EOG studies show that the Bighead Carp also detects PGFs well (Lim and Sorensen, in preparation). Second, seemingly identical to the Goldfish, the Silver Carp olfactory system is quickly and specifically masculinized to PGFs using MT treatment and only these fish respond behaviorally, strongly suggesting that, as established in the Goldfish, PGFs are indeed sex pheromones [13,18]. Of course, it is important to confirm sexual dimorphism in wild-caught sexually mature male and female Silver Carps as has been done for Eurasian Carps because while MT treatment stimulates normal courtship (and presumably attraction), differences in other aspects of behavior could exist [15,18].

As well established in the Eurasian Carps [11–13], $PGF_{2\alpha}$-treatment clearly evokes the production of a species-specific PGF-based sex pheromone in the Silver Carp. However, unlike for the Common Carp [11,12], this blend seems to drive pheromonal activity in the Silver Carp. This may not be unreasonable because pheromonal blends are important in many sympatric species including many insects [29,30]. Notably, the Silver Carp seemingly evolved in sympatry with other carps including the Bighead Carp, while the Common Carp and Goldfish did not [21], so Bigheaded Carps may have needed to evolve more precise species-isolation mechanisms to mediate reproductive behavior. The PGF pheromone is likely specifically associated with the act of spawning as in the Eurasian Carps. Because immature Bighead and Silver readily intermix [31], common body odors probably do not serve a species-isolation function as in Eurasian carps [30]. Sex pheromonally-based isolation mechanisms in Bigheaded Carps cannot be perfect given the observation of hybrid Silver-Bighead Carp in North America but not in China. Perhaps natural habitat features may help completely isolate these species when spawning in China, complimenting the actions of the pheromone [32].

Our study has several notable strengths and a few weaknesses. Using well established electrophysiological and endocrinological techniques we clearly show for the first time that a relatively unstudied and important species, the Silver Carp, detects PGFs with high sensitivity using multiple receptors, and that this sensitivity is androgen dependent. Using two different behavioral tests, one of which has been used many times,

we also show that androgen-treated carp respond to these PGFs, whose release we also show can be induced using established implant techniques that also stimulate release distinctive odors and mixtures of PGFs. The strongest behavioral responses were clearly seen to the blends of PGFs released by Silver Carp which also would have, been exposed to largely undiluted odors streams in the rectangular tank, suggesting a failure to detect 15K-PGF$_{2\alpha}$ in the Bigheaded Carp odor was unlikely to be the cause of its mixture's lack of potency. Nevertheless, while these laboratory techniques and assays have been shown to mimic behaviors and release in a closely-related model cyprinid, the Goldfish, there is always a possibility Silver Carp may be different. Additionally, this study measured slightly different ratios of PGFs released by juvenile Common Carp than we saw in previous work using much larger individuals [11], suggesting more study of PGF metabolism and release in fish is needed to fully understand the causes and significance of PGF metabolism. Accordingly, specific PGF ratios measured by small implanted laboratory fishes should be interpreted with caution. Whether or not 15K-PGF$_{2\alpha}$ and dh15K-PGF$_{2\alpha}$ play different roles in the Silver Carp pheromone must also be examined in greater detail, ideally with naturally mature and ovulated Silver Carp. Ideally larger, more natural test arenas or ponds might be used too and many more individuals than we were able to manage in this multiple-year laboratory-based study that was limited by ability to treat more than a few dozen fish with androgen for extended times. Nevertheless, given the fact that PGF$_{2\alpha}$ production, metabolism, and release is now known to be intimately associated with both ovulation and the expression of female sexual behavior (they increase within minutes of each other in the Goldfish [18]), and the fact that the Goldfish has proven to be a powerful and reasonable model of many cyprinids [5,6], it seems highly likely that the Bigheaded Carps use also PGFs as female spawning pheromone as seen in many other cyprinids.

In conclusion, while our study, a relatively simple test of PGF sensitivity, release, and responses in the laboratory, cannot be considered definitive, when taken as a whole it appears to confirm that the Silver Carp and presumably its congener the Bighead Carp use PGF-based sex pheromones to attract each other and mediate spawning. These carps thus resemble dozens of other fish species, including the invasive Common Carp, in their use of this class of pheromones, opening new doors for invasive carp management. The seeming potency and specificity of their pheromone signal(s) are especially notable. A PGF-pheromone in the Silver Carp might have several uses. For example, given its high potency, it could be easily used as an attractant to bring males to specific locations for removal or to be counted [9]. By including feeding attractants outside of the mating season [33–35], an attractant strategy using different types of attractants including sex pheromones might be especially valuable [9]. It seems especially notable that PGF$_{2\alpha}$-implanted Silver Carps could be used an inexpensive platform to produce and release pheromones in wild fishes, as has been shown in the invasive Common Carp [12]. Additionally, because hormonal PGF$_{2\alpha}$ is known to function as a female reproductive hormone that drives female behavior [14], it creates the opportunity to use PGF$_{2\alpha}$-implanted carps as Judas fish, which can lead managers to shoals of mature fish [34,35]. Lastly, recent studies have shown that PGF can be measured in waterways, making it possible to track PGFs together with eDNA to survey for Bigheaded Carps, while also identifying Carp gender and health [9,35]. Other studies now hint that Bigheaded Carps also use sex steroids as priming pheromones (Lim and Sorensen, in preparation) similar to the Goldfish [5,6,10], suggesting a reliance on sex pheromones for many aspects of reproduction. We hope our study will serve as a foundation for future studies of pheromones in wild, free-ranging fish that will both examine and develop the basic science and application of sex pheromones in the Bigheaded Carps.

## 4. Materials and Methods

### 4.1. Fish

Juvenile Silver and Bighead Carp (10–15 cm in total length, sex undeterminable) were obtained from the United States Geological Survey Columbia Environmental Research Center (Columbia, MO, USA) and housed in 450-L flow-through, circular tanks supplied with 21 °C well water on a 18 h:8 h L:D photoperiod. These fish were fed daily on a diet consisting primarily of spirulina powder and

chlorella [31]. Juvenile Common Carp were obtained from Osage Catfisheries (Osage Beach, MO, USA) and kept under the same conditions, except they were fed food pellets (Silvercup, Salt Lake City, UT, USA). For electro-olfactogram recording (EOG) and behavioral experiments, one hundred Silver Carp were treated with methyltestosterone (MT, a synthetic and affordable androgenic steroid) by adding 232 $\mu$L of a $10^{-2}$ Molar (M) solution of MT in ethanol to holding tanks once a day (for over a year) to create a concentration of approximately $10^{-8}$ M. As a control, a dozen Silver Carp were exposed to ethanol alone. This procedure was one that we [17] and others [16] have used previously, which shows that MT can completely masculinize the olfactory systems of several female cyprinid fishes, including the Goldfish, which are otherwise behaviorally unresponsive to waterborne pheromonal PGFs. Experimental holding conditions and methods were approved by the University of Minnesota Institutional Animal Use and Care Committee protocol #1009A88894.

### 4.2. Determining Olfactory Sensitivity of Untreated and Masculinized Silver Carps to Prostaglandins

Olfactory sensitivity in Silver Carp was discerned using electro-olfactogram recording (EOG), an extracellular recording technique that accurately assesses olfactory organ sensitivity following well established protocols [13,14,17,18]. Briefly, fish were anesthetized using 0.01% MS-222 (Western Chemical Inc., Ferndale, WA, USA), placed on a recording stand, which irrigated their gills with 0.001% MS-222, after which the skin flap protecting their nares was carefully removed, while a recording electrode was placed over their olfactory epithelium and an indifferent electrode was placed on their skin. Odors were then pulsed for 5-s with 2-min inter-stimulus intervals across their nose, DC voltage transients were measured, and expressed relative to our standard, $10^{-5}$ M L-Serine (a food odor we have used in the past). All stimuli, including well water control, were tested at least twice (more if responses differed) and responses were averaged. In addition to three PGEs, three PGFs were tested: $PGF_{2\alpha}$, 15keto-$PGF_{2\alpha}$ (15K-$PGF_{2\alpha}$), and 13,14-dihydro-15keto-$PGF_{2\alpha}$ (dh15K-$PGF_{2\alpha}$), the primary metabolites of $PGF_{2\alpha}$ in the Common Carp as well as the Silver Carp (see section below). In each case, we tested PGF concentrations ranging from $10^{-12}$ M to $10^{-8}$ M, starting with the lowest concentration and ending with the highest and then the standard. Blank control responses (if measured) were subtracted form test responses, and then divided by the standard. The PGFs were of the highest quality (98%) and obtained from Cayman Chemicals (Ann Arbor, MI, USA), while the L-serine came from Sigma Co (99% pure; St Louis, MO, USA). All stimuli were made fresh each day. We tested groups of 3–6 untreated Silver Carp, Silver Carp that had been treated with ethanol control for 3 months and Silver Carp that had been treated with MT (masculinized) for 3 months because we found the complete masculinization of the olfactory system within 6 weeks of MT exposure (or implantation) in closely related Goldfish [17]. Means and 95% confidence intervals were calculated and plotted. In addition to these experiments, three experiments were conducted to determine if the three PGFs might be discerned by different olfactory receptor systems. These experiments calculated a mixture discrimination index (MDI) following established procedures [36]. The MDI is calculated by determining EOG responses to an equipotent mixture of two compounds of interest and then EOG responses to each component at twice the concentration used in the mixture. The index is then calculated by dividing the response to the mixture by the total of half that of the responses to each compound. A score of 1.0 is predicted if the two compounds are detected by the same olfactory receptor and differences from 1.0 are calculated by paired *t*-tests ($p < 0.05$).

### 4.3. Initial Tests of the Behavioral Activity of Individual PGFs

The attractiveness of individual PGFs to Silver Carp was tested in a circular maze we have used before to test attraction in both Goldfish and Common Carp [11,12,37]. Following established protocols, groups of 5 Silver Carp were introduced into this circular tank (1.5 m diameter) divided into two semi-circular sections connected by a central drained area. Overhead infrared lights and cameras allowed for observation of fish from a monitor so that trials were conducted in near complete darkness. After a 1 h acclimation period, odor stimuli were injected (10 mL/min) via silicon tubing at opposite

ends of the semi-circular sections of the tank using a peristaltic pump. Dye tests showed that this rate of release contained odors to their respective side of the maze for at least a 15-min period. Each trial consisted of a pre-test, control test, and PGF odor test. During the pre-test, well water control from the same source was pumped into both semi-circular sections. The pre-test was immediately followed by a control test during which a second preparation of blank well water was pumped into the side of the tank where fish were observed the least during the pre-test. The control test was followed by a PGF odor test during which time $10^{-8}$ M of a PGF (Cayman Chemicals Co. Ann Arbor, MI, USA) was pumped into the side with the fewest observations of fish during the control test. Pre-test, control test and odor tests were all 15 min long with a 3–4 min interval between subsequent tests. During test periods, observations of the position of each fish in the tank were taken at 15-s intervals for a total of 300 observations per test. Twelve trials were conducted for each stimulus. A relative attraction value was calculated by summing the total number of observations on the side where the odor was released (or blank for control tests), dividing it by the total number of observations and subtracting the percentage of observations on the same side during the preceding test to yield relative attraction. If fish did not move, trials were not used. One-tailed paired *t*-tests were used to make comparisons between test and pre-tests after removing possible outliers following Winsorization procedure in which the highest and lowest values were removed.

### 4.4. Determining the Mixtures of PGFs Released by PGF$_{2\alpha}$-Implanted Bigheaded Carps

Because we did not have access to ovulated carps to test whether and what natural mixtures of PGFs might function as a sex pheromone, we implanted fish with PGF$_{2\alpha}$, a technique that we knew to simulate natural pheromone release in the Common Carp [11,12]. We followed the protocols developed for the Common Carp for this purpose except we only had immature fish to use [11,12]. Briefly, 9 juvenile Silver, Bighead, and Common Carp (all approximately 50 g ($\pm$10 g); gonadosomatic index < 1%) were anesthetized with 0.01% MS-222, and an osmotic pump (model 2ML1; Alzet, Durect Co., Cupertino, CA, USA) containing 1 mg of PGF$_{2\alpha}$ in sterile water was inserted into each individual's body cavity through an 1 cm incision, which was then sutured. This dosage of PGF$_{2\alpha}$ was designed to be released at about 1 µg/g body weight for 8 days following implantation. Fish were held in 150 L flowing water tanks and then on days 0, 1, 3, 5, 7, and 9 placed into 7.5 L of well water as groups of 3 for 1 h, after which multiple 1-L aliquots were collected. Several samples were immediately frozen ($-80\ ^\circ$C) for behavioral tests (see below), while other samples were kept for PGF analysis. Samples from day three were used in this study because that was when peak PGF release has been noted. Samples of holding water were then extracted to obtain their nonpolar compounds including PGFs by passing it through an activated C18 solid-phase extraction (SPE) cartridge (Waters, Milford, MA, USA), which was then eluted with 5 mL of HPLC-grade methanol (Sigma-Aldrich, St. Louis, MO, USA) and dried down under a stream of nitrogen before being reconstituted in a 1 mL mixture of methanol and distilled water (50:50 *v/v*).

For identification and quantification, 50 µL of the reconstituted eluate was then injected into an Eclipse XDB C18 RS column (250 × 4.6 mm, particle size 5 µm; Agilent Technologies, Santa Clara, CA, USA) following Lim and Sorensen [12]. The LC system allowed us to separate PGFs and was interfaced directly with an ion-trap mass spectrometer (LCQ-1, ThermoScientific, Waltham, MA, USA) equipped with an electrospray ionization source (ion trap operated in the negative ion mode with a spray voltage of 5 kV; sheath gas was 99% pure nitrogen at 60 psi; sheath fluid was 50:50 20 mM triethylamine: acetonitrile (*v/v*)). Chromatographic separation and ionization patterns were compared with those of standards for PGF$_{2\alpha}$, 15K-PGF$_{2\alpha}$, and dh15K-PGF$_{2\alpha}$ (Cayman Chemical, MI, USA). The flow rate of the LC was 1 mL/min and the mobile phase consisted of 0.1% formic acid in acetonitrile (A) and 0.1% formic acid in distilled water (B). LC separation was carried out with a linear solvent gradient program of 35% A to 65% B at 0 min, 65% A to 35% B at 12 min, 35% A to 65% B at 14 min, and 35% A to 65% B at 18 min. To quantify these compounds, we used selected ion monitoring. A five-point linear calibration curve was established by extracting Carp odor spiked with PGF$_{2\alpha}$

(1–100 μg/mL) and 15K-PGF$_{2\alpha}$ and dh15K-PGF$_{2\alpha}$ (10–1000 ng/mL) as employed previously [11]. Daughter ions of 291, 309, 317, and 335 *m/z* were monitored for PGF$_{2\alpha}$ and dh15K-PGF$_{2\alpha}$, 351 and 333 *m/z* for 15K-PGF$_{2\alpha}$. An example mass spectrum of 15K-PGF$_{2\alpha}$ has already been published [11]. Data acquisition and analysis were carried out with the Xcalibur (v. 2.0.7; Thermo Fisher Scientific, MA, USA). Relative release rates were tabulated by weight and molarity species (ng/h/g) and these ratios were compared using chi-square analysis ($p < 0.05$).

### 4.5. Determining Whether the Odor of PGF$_{2\alpha}$-Implanted Carp Functions as a Species-Specific Behavioral Attractant

We deigned a new type of rectangular tank for this experiment which included flow during pre-test periods, which seemed to facilitate more natural behaviors including shoaling. Briefly, a large (65 cm (W) × 265 cm (L) × 45 cm (D); 780 L), dimly-lit (5 w bulb) rectangular tank filled with 18 °C well water was used to assess attraction of Silver Carp to PGF$_{2\alpha}$-implanted fish water (Figure 5). This tank had a darkened window (shielded by a curtain) on the front across which a vertical line had been drawn so we could note fish position on each half of the tank. Odor injection tubes were placed on each corner of the tank through which odors could be pumped at a rate of 10 mL/min through silicon tubing using a peristaltic pump. Dye tests showed that odors added to the tank in this manner did not enter the other side of the tank for at least 15 min and that fish were generally exposed to relatively undiluted stimuli. We tested responses of groups of 4 masculinized Silver Carp (to simulate natural groupings [31]). These carp had been masculinized for 10 months, in excess of the 4 months found necessary for female Goldfish to exhibit normal male courtship behaviors [18], as pilot tests showed that untreated juvenile Silver Carp were not behaviorally responsive to PGFs (Figure 2). We tested the effects of exposing masculinized Silver Carp to holding waters of PGF$_{2\alpha}$-implanted Silver Carp, Bighead Carp, and Common Carp, which were obtained by holding them in a 18 °C water bath for 4 h (we used waters collected on the third day after implantation when PGF release rates were maximal (unpublished results)). For each trial, a group of 4 Silver Carp was introduced to the assay tank and allowed to acclimate for 80 min while 18 °C well water was slowly introduced (5 L/min) into the tank, after which it was turned off. Three sequential experimental periods then followed during which time fish positions were monitored in each half of the tank by an observer unaware of the test odor: (1) a 10-min pre-test during which time nothing was added to the tank; (2) a 10-min control period during which the well water control was added to the side of the test tank seen to have the fewest fish and nothing to other (a test for the effects of water addition and passage of time); (3) a 10-min test period during which time the test odor was added to the side of the tank determined to have the fewest fish while well water control was added to the other side. During these test-periods, the number of fish on either the right or left side of the tank was noted every 15 s and then summarized over the 10-min period. There was no obvious bias in the tank ($p > 0.05$). Relative attraction was calculated by determining distribution, i.e., the number of time-points fish spent on the side that the test odor was added to (total number of fish seen summated across all 40 15-s periods divided by 160 (the total number of observations) × 100). We then analyzed changes in fish distribution using paired *t*-tests ($p < 0.05$, 1 tailed as we analyzing attraction only) (Graphpad Prism, CA, USA) after confirming normal distribution of data. One trial was conducted per day, and fish were not re-used so we were able to perform 7 trials (control and experimental) for each stimulus. This approach normalizes data and incorporates possible inherent side biases in the tank, a technique that has been used in previous studies [11,12,31] including the circular maze work.

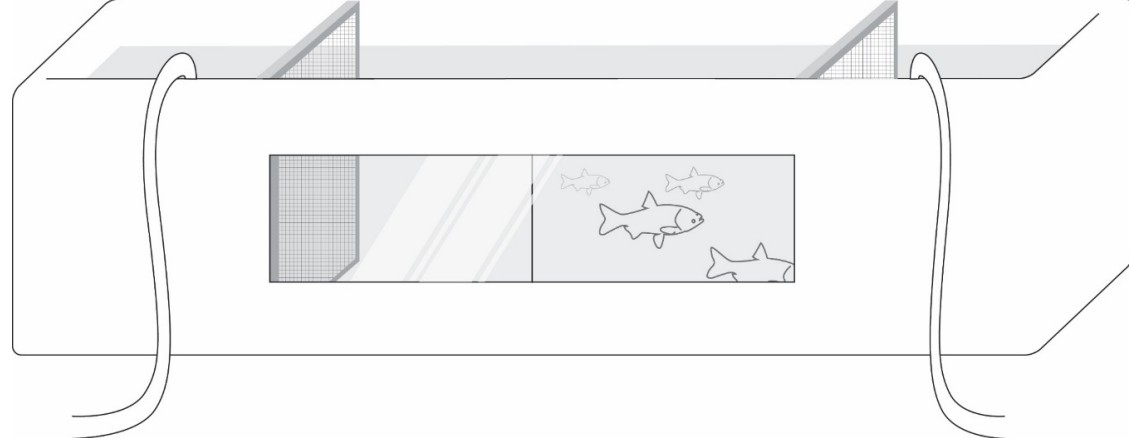

**Figure 5.** Rectangular test tank used for testing the behavioral activity of $PGF_{2\alpha}$-implanted carp odors showing its observation window and the odor injection tubes. The tank measured 2.6 m in length.

*4.6. Determining Whether the PGFs Released by $PGF_{2\alpha}$-Implanted Carp Function as a Species-Specific Attractant*

The behavioral activity of PGFs in holding waters was tested in a final experiment that used the same protocol and rectangular tank as tests using $PGF_{2\alpha}$-implanted carp odor except that the PGFs were tested alone. We tested the effects of exposing groups of 4 masculinized Silver Carp to the mixture of PGFs found in Silver Carp holding water ($PGF_{2\alpha}$:15K-$PGF_{2\alpha}$:dh15K-$PGF_{2\alpha}$ 94.7:2.3:2.7) and Bighead Carp holding water (99.0:0.1:0.8, which did not differ statistically from that of Common Carp holding water) after giving these fish at least 3 weeks to recover from the previous experiment (n = 7 groups for each). During this 3-week recovery period, fish were re-assembled and held as one large group, before being subdivided again in a random fashion for reuse (a procedure we have used successfully in the past). Total concentration of all PGFs added at a rate of 10 mL/min was $2.55 \times 10^{-8}$ M, mimicking that in holding water. Final total concentration in the experimental side of test tank (assuming complete mixing) would have been about 1/1000 of this value because of dilution, however because of the way the odor was added and way fish swam at the surface near the odor inlet, they would typically have been exposed to only slightly diluted odors. Data were analyzed as for the $PGF_{2\alpha}$-implanted odor tests. All behavioral testing procedures used in this study were also approved by the Institutional Animal Care and Use Committee of the University of Minnesota protocol #1009A88894.

**Supplementary Materials:** The following are available online at http://www.mdpi.com/2410-3888/4/2/27/s1, Figure S1: EOG responses of juvenile Silver Carp to nanomolar concentrations of PGEs and PGFs.

**Author Contributions:** P.W.S. conceived of the study, procured the funding, directed the research, and wrote the manuscript. J.M.L. conducted the circular maze experiments. R.G. designed the behavior experiments in the rectangular tank which M.C.P.R. then performed. H.L. conducted and analyzed the PGF release experiments. All authors edited the final manuscript.

**Funding:** This work was funded by the U.S. Geological Survey (Columbia, MO, USA).

**Acknowledgments:** We thank Liz Fox for technical help with the EOG, Dan Krause for his help with behavior tank setup, Rosie Daniels for her help with the figures, Robin Calfee for help with the fish, and Ed Little for his advice.

**Conflicts of Interest:** The authors declare no conflict of interest.

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
