# Peer review of "A Blend of F Prostaglandins Functions as an Attractive Sex Pheromone in Silver Carp"

_fishes, doi:10.3390/fishes4020027_

Round 1
Reviewer 1 Report
Sorensen et al. describe a series of experiments focused on the potential role of F-series prostaglandins as species-specific sex pheromones in silver carp. The authors used electroolfactogram recordings to determine olfactory responses to PGFs, LCMS to quantify PGFs in holding water, and behavioral assays to evaluate PGF mixtures as species-specific pheromones, and conclude that silver carp likely use a blend of PGFs as species-specific sex pheromones.
The results are potentially quite interesting; although many fish are known to respond to specific-specific odors, this would be, to my knowledge, the first direct evidence that the ratio of known components confers species-specificity. Likewise, fishery managers may be interested in the potential to use an understanding of pheromone communication in Asian carps for control purposes.
However, I have concerns with the experiments and am not convinced they offer conclusive support for the hypothesis that a blend of PGFs acts as a species-specific pheromone in silver carp. Rather, the data support the conclusion that PGF-implanted juvenile silver carp release a species-specific odor comprising at least one of the tested prostaglandins.
Major concerns:
1) The experiments indicate that at least one of the three PGFs acts as a sex pheromone in silver carp, but do not directly test the importance of the mixture or any individual component. Lim and Sorensen (2011) report that common carp respond to a mixture of these three PGFs, but, using tests of each individual compound, found that only PGF2α is behaviorally active. Therefore, the activity of the mixture might be due to one of its components not the complete blend. For example, silver carp may use only 15k-PGF2α as a pheromone and not respond to the bighead carp mixture because the concentration of 15k-PGF2α is too low, not because the ratio of the three components is important. Based on the data in Table 1, the concentration of 15k-PGF2α in the behavior tests would have been roughly 10-14M for the bighead carp trials, and several orders of magnitude higher for the silver carp trials.
2) The release rates and the ratios of the three PGFs for the implanted juvenile common carp reported here are considerably different that those for naturally ovulated female common carp reported by Lim and Sorensen 2011 (PGF2α = 3.43, 15k-PGF2α = 5.67, dh15K-PGF2α = 2.27 ng/g/h). What justifies use of implanted juveniles as a proxy to study ovulated female sex pheromones, especially when the ratios, the focus of the study, are not comparable?
3) The ratios in the text appear to be incorrect based on the data in Table 1. For example, bighead carp release 630 ng of PGF2α and 0.6 ng 15k-PGF2α, a difference of 1000x. These two compounds have the same molecular mass, so how is the molar ratio 99:1? Likewise, no ratios are reported for the behavior section, so it is unclear if the behavior tests used appropriate ratios.
Line 2: The title is too strong, and not supported by the data.
Line 46: Consider revising to “produced as a hormone by ovulated…”.
Line 58: The novelty of the current study would be clearer if it were noted that the ratios of known PGFs do not mediate species-specificity in goldfish and common carp.
Line 64: Insert comma after [18].
Line 89: include exact p-values, test statistics, and df throughout the results.
Lines 90-105: Statistical tests more formal than comparing confidence intervals should be used for comparing across treatments and determining the detection threshold.
Line 115: Using Chi-Square to test for differences in ratio seems inappropriate. A more robust comparison would include the among-individual variation.
Line 118: The ratios are wrong based on the data in Table 1. For example, bighead carp release 630 ng of PGF2α and 0.6 ng 15k-PGF2α, a difference of 1000x. These two compounds have the same molecular mass, so how is the molar ratio 99:1?
Line 217: “this procedure was one”
Lines 222-242: Were responses to blanks subtracted from responses to Ser and PGs?
Line 232: should be “three” not “Three”.
Line 236: Instead of “the PGFs were of highest quality”, report the purity.
Line 255: I think the authors mean samples from day 3, not three day old samples.
Line 261-277: I am not a chemist, but my understanding is that these methods are not able to discriminate between the many possible PG structures. In fact, Lim and Sorensen 2011 specifically state that these methods cannot differentiate between PGF2α and dh15K-PGF2α. The authors should acknowledge the limitations of the method and be clear on what exactly can be concluded based on these data.
Line 291: insert comma after silver carp
Line 340: the time fish spent on each side was not recorded, only the number of timepoints when fish were observed on a side.
Line 315-325: What were the specific ratios tested? Table 1 and the associated text are conflicting, and no ratios are reported here.
Line 322: It seems odd to keep the total concentration the same rather than keeping one compound at the same concentration and then adjusting the concentration of the other compounds to manipulate the ratio.
Reviewer 2 Report
The study by Peter Sorensen and colleagues tested if the Asian Silver Carp uses a F2α-prostaglandin-type sex pheromone to attract conspecifics. The study is generally well written and presented and adds to the growing body of evidence that prostaglandin-derived metabolites have an ubiquitous function as female sex pheromones in Cyprinids. However, some methodological concerns should be better addressed throughout the manuscript text and alternative explanations for the results observed could be considered. Language style can also be improved in some places of the manuscript.
Major concerns
1. Electroolfactogram (EOG) recordings.
The authors conclude from their analyses that the olfactory responses to PGF2α (Fig. 1A) are larger in MT-treated fish than in non-treated and EtOH-treated fish, yet the graph does not show that very well (overlapping 95% confidence intervals). The authors should present additional statistical analyses to proof this difference or reconsider their respective conclusion. Then, the authors could have tested the three PGFs at higher concentrations (e.g. 10-7M and 10-6M) to obtain more complete concentration-response curves and conclude on receptor saturation which they later-on discuss (e.g. in other teleosts such as the Mozambique tilapia receptor saturation to its sex pheromone is reached only at about 10-6M (Keller-Costa et al., 2014 CurrBiol or JExpBiol)). Also, in their discussion the authors suggest independent receptor mechanisms for PGF2α and 15KT-PFG2α in Silver Carp, however, for this, cross-adaptation tests should have been performed to understand if the three PGFs act on the same olfactory receptor or if they can indeed be discerned by the fish based on different receptor mechanisms. The authors should either provide additional data on these matters or adapt their discussion acknowledging these issues and soften their conclusions.
2. Chemical analysis of holding water samples
This section would become more interesting to the readership if the authors would show the representative LC-MS chemical profiles of holding water samples for each of the three implanted carp species. In addition, it would be informative to see a profile of the standard mixture of the three PGFs that was used to quantify the same as well as a chemical profile of non-implanted (control) fish.
3. Behavioural experiments
It can be informative for the readers to also see the behavioural response of untreated, juvenile carp to the PGF mixtures (in their methods section, the authors mention that this was tested but results are not presented). Also, in their discussion, the authors suggest that the Silver Carp sex pheromone is a blend of PGF2α compounds. However, the authors did neither test nor discuss the possible effect of each PGF on its own. Couldn’t it be also possible that 15KT-PGF2α is the major driver of activity, given that EOG recordings revealed highest sensitivity to this compound and that chemical analysis detected at least 10 times more 15K-PGF2α release in Silver Carp than in Bighead Carp and Common Carp? If the artificial Silver Carp-based PGF2α-mixture used in figure 3A contained accordingly more 15K-PGF2α than this might as well explain the observed behavioural responses in 3A vs 3B? This alternative viewpoint could be considered by the authors in their discussion and interpretation of results. Following this, the authors should also provide the precise composition of the PGF mixtures, giving the exact quantity of each PGF in each mix.
Specific comments
Abstract
Line 17: I would suggest replacing the term “spawning pheromone by the more general term “sex pheromone”. The study does not show the effect of F2α prostaglandins on actual spawning, but on attraction. The word spawning pheromone may mislead the reader`s expectations. Consider changing this throughout the manuscript text.
Introduction
Page 1, Line 32: Pheromones are not restricted to animals, consider replacing “animals” by e.g. “living organisms”
Page 1, Line 39: “…, candidates for use attractants for invasive fishes…” Unclear, please rephrase/correct language.
Page 1: The state of the art on sex pheromones in teleost fishes presented by the authors is very much restricted to Cyprinids and the author’s own previous publications. However, Cyprinids are not the only teleost fishes that have had their sex pheromones investigated. The authors could present a more balanced literature overview here by referring also to other teleost models, e.g. cichlids and gobies or catfish.
Page 2, Line 56: “…and these metabolites” replace by “…,15K-PGF2α and dh15K-PGF2α”
Page 2, Line 62: delete “these”
Page 2, Line 79: should read “…and is difficult to study in the field”
Page 2, Lines 80-81: Please include a reference for the study of Common Carp.
Results
Page 3, Lines 95-98: Looking at figure 1A, it is not clear to me that the olfactory response of MT treated Silver Carp to PGF2α is indeed larger than the responses of non-treated and EtOH treated fish since the 95% confidence intervals overlap (btw, why showing 95% confidence intervals and not standard errors?). The authors should either run a statistical test to proof the existence of the difference claimed here or reconsider the interpretation of this result.
Page 4 Lines 122: correct “…implanted specifics” with “…implanted conspecifics…”?
Page 5, Line 163: replace “washings” by “holding water”?
Figure 2
Page 5: Is the figure showing mean values and standard errors? Please state what is shown in the figure legend.
Figure Legend 3
Page 6, Line 144: it should read “Two test stimuli…”
Discussion
Page 6, Line 150: Please replace “spawning pheromone by “sex pheromone” (see my earlier comment on this). Please change throughout the discussion.
Page 6, Lines 152-154: Please rephrase/revise language.
Page 6, Lines 156: Please revise language.
Page 6,7: Lines 167-170: Check language. This sentence is very long, consider dividing it into two. Also, cross-adaptation studies would be more effective to check for independent receptor mechanisms than looking at concentration-response curve profiles alone (btw you are presenting concentration-response curves not dose-response curves).
Page 7, Line 171: It should read “Bigheaded carps” and “secondary messengers”
Line 172: Please rephrase/revise language. It is unclear what is meant by “definitive studies of species-specificity”?
Line 177: It should read “…as it has been done for Eurasian Carps”?
Line 180: Replace “pheromonal specificity” by “pheromonal activity”?
Lines 181-185: Consider the possibility that instead of a PGF blend, 15K-PGF2a alone might suffice to trigger attraction? (Please see my earlier comment).
Line 187: Check language. It should read “…to be a definitive test…”
Line 189: Consider replacing “mediate spawning” by “attract conspecifics” since this is what you show?
Line 191: Replace “Bigheaded carp” by “Silver Carp”?
Lines 195-197: This sentence is odd. Please rephrase/correct language.
Methods
Page 7, Line 212: What type of pellets?
Pages 7 and 8; Lines 217-221: Rephrase / check language.
Page 8, Paragraph 4.2. Please state how many replicate fish were in each treatment group (i.e. non-treated, methanol-treated, MT-treated) that were subjected to EOG recordings.
Line 256: Please insert “Holding water” before “Samples were then extracted…”
Line 257: Please replace “C18 column” by “C18 solid-phase extraction (SPE) cartridge”
Line 261: It should read “eluate was injected”
Page 9, Lines 289-290: It would be nice to see these unpublished results to be included in Figure 3.
Line 292: Replace “thawed” by “obtained”?
Line 298: Replace “Control” by “control”
Line 299: Please replace “time blank” by “the” and “and nothing to other” by e.g. “while no stimulus was added to the other side of the tank”
Did the fish show any preference for one tank side or the other during the pre-test and control periods? I.e. was there one site at which stimuli were more often introduced or was stimuli introduction even at both sites? Please state how often test stimuli were delivered at each side.
Were holding water donor carp kept separately from the behavioural assay performers? Or could odor-based recognition of familiar tank-mates have occurred during the behavioural assays?
Line 300: Replace “Test” by “test”
Line 307: Change into “…as we are looking at...”)?
Line 308: Change into “after confirming normal distribution of the data”?
Page 10, Line 321: Please state that it is “Silver Carp holding water and Bighead Carp holding water” and revise the English of the text written within brackets.
If I understood correctly, the Silver Carp that performed the assay in Fig. 2 were re-used for the PGF-mixture tests in Fig. 3. Were the groups re-assembled for this new series of tests or was group composition kept constant (i.e. the same 4 individuals performed the behavioural tests together twice)? Did all assay-performing fish stay together in the same tank during the 3-weeks recovery period? Please clarify.
Line 322-235: Please give the precise composition of the two artificial PGF2α mixtures used in Figure 3, i.e. exact quantity of each PGF in each mix.
Supplemental Figure S1
Title: Please rephrase “F2 series have function” into e.g. “F2 prostaglandins are active”
Ethical Statement
Please provide an ethical statement on the experimental use of vertebrates in this study.
Author Response
Please see attached PDF reviewer 2

Reviewer 3 Report
This manuscript advances the so needed knowledge about carp pheromones. Although the manuscript is good for what it intends, a proof of content, I find the lack of replicates make this work weak. It is for that reason that I suggest to change the title to “Species-specific blend of F prostaglandins functions as a putative attractive pheromone in Silver Carp”. As noted in the discussion, there is still work to do to describe pheromone communication in this species.
The figures on my manuscript version are pixelated and have bad quality. If these are the original figures, the authors should improve the quality of the graphs.
Minor Comments
Figure 1. Change X axis to “Relative EOG Response”.
Supplementary Fig. 1 is very sloppy. Axes need to be properly explained (i.e. Relative EOG Responses (% Serine), a legend for the prostaglandin forms…). There is no N or standard error values. Write the title of the graph and the legend properly (e.g. 10-9 M not -9M).
Add space between numbers and units.
Add space between numbers and symbol equal.
Line 91. Sexually-dimorphic is generally related with differences between male and female. Although the use of this term is probably not wrong in this context, it made reading confusing. I suggest to use the terms sex maturation or implanted fish or similar.
Line 116. Put space between X2 and = and 35.2. I doubt that a χ2 analysis with a sample size of 3 is correct. I believe an exact text will be more appropriate. Please change the analysis.
Line 170. put a citation for that sentence. There is other reasons why there is not saturation in the EOG curves (e.g. differences in the number of receptors). Please cite other points of view and discuss how can help to understand pheromone research in carp.
Line 196. Correct typo " the entire signal n relevant context".
The full paragraph from line 194 to 200 is confusing. I suggest rewriting.
Bibliography have format errors.
Round 2
Reviewer 1 Report
Sorensen et al., submit a revised version of fishes- 426683, now titled "A blend of F prostaglandins functions as an attractive 2 sex pheromone in Silver Carp". The authors generally do a good job responding to my comments, and the manuscript is improved.
However, I still have some concerns. First, I am not satisfied with the author's response to my request for specific p-values, test statistics, and df. I am OK if the authors do not include p-values and test statistics for nonsignificant results, but these tests do need to be paired with number of replicates so readers can evaluate power. For significant results, I disagree that the details of the tests (specific p-value, etc) make the manuscript overly long or difficult to follow.
I also have concerns with other aspects of the statistics. 1) I could find no information about how many replicates were conducted for the new experiment that was added to the revision. 2) I’m skeptical about the removal of outliers (line 351), especially if the sample size for this test was similar to the others (n=7). If nothing else, the authors should include details of what outliers were removed. 3) The authors suggest that confidence intervals are appropriate statistical evaluations for their EOG data because they only intended to determine if a compound was active. If this is the case, they should remove the text where they compare detection thresholds (e.g. line 107). 4) I am also skeptical of doing t-tests on EOG data with n = 3; I am not sure it is appropriate to do statistics with only 3 replicates. I see in the methods that 3-6 replicates were done for EOGs, so maybe these statistical tests had 6 replicates (figure 1 indicates n =3, so I think not); again, it would be useful to have this information included with results.
Second, I think the authors should be more direct in their discussion of the use of implanted juveniles. I appreciate that using adults is logistically challenging, and agree that the data are interesting even though not collected from adults. However, I think the authors should be very clear about this weakness. For example, the authors mention this weakness at line 250 and call for studies on PGF metabolism, but this would be a good spot to also caution readers that the reported release rates/ratios may not be ecologically relevant.
Author Response
March 27 2019
Sorensen et al., submit a revised version of fishes- 426683, now titled "A blend of F prostaglandins functions as an attractive 2 sex pheromone in Silver Carp". The authors generally do a good job responding to my comments, and the manuscript is improved.
-Thank you; we think it is improved too.
However, I still have some concerns. First, I am not satisfied with the author's response to my request for specific p-values, test statistics, and df. I am OK if the authors do not include p-values and test statistics for nonsignificant results, but these tests do need to be paired with number of replicates so readers can evaluate power. For significant results, I disagree that the details of the tests (specific p-value, etc) make the manuscript overly long or difficult to follow.
-It is not unreasonable to ask for exact p values for the tests which proved significant. The Results section has been edited to include these values. We added df at the same time.
I also have concerns with other aspects of the statistics. 1) I could find no information about how many replicates were conducted for the new experiment that was added to the revision. 2) I’m skeptical about the removal of outliers (line 351), especially if the sample size for this test was similar to the others (n=7). If nothing else, the authors should include details of what outliers were removed.
-Twelve trials were conducted for each of these "new" experiments (i.e.more). This value is now inserted in the figure legend and methods. Winsorization involves removing both the highest and lowest values from data sets so is very conservative. This is now described as well as the df (9) in the edited Results and methods.
3) The authors suggest that confidence intervals are appropriate statistical evaluations for their EOG data because they only intended to determine if a compound was active. If this is the case, they should remove the text where they compare detection thresholds (e.g. line 107). 4) I am also skeptical of doing t-tests on EOG data with n = 3; I am not sure it is appropriate to do statistics with only 3 replicates. I see in the methods that 3-6 replicates were done for EOGs, so maybe these statistical tests had 6 replicates (figure 1 indicates n =3, so I think not); again, it would be useful to have this information included with results.
-I think the reviewer thought we were comparing detection thresholds (which would not have been valid) when we were actually comparing maximal responses elicited by 10-8M concentrations of PGF. This section was edited to clarify this important issue. To further emphasize the credibility of these differences in EOG responsiveness we added another dataset (Supplementary Figure 2) to the Supplementary Data which shows even larger EOG responses to PGFs after a year of MT treatment (n=5 in this case). (In an attempt to keep the manuscript short we had regrettably not included much of what now appears to be key supporting data in the original draft).
Second, I think the authors should be more direct in their discussion of the use of implanted juveniles. I appreciate that using adults is logistically challenging, and agree that the data are interesting even though not collected from adults. However, I think the authors should be very clear about this weakness. For example, the authors mention this weakness at line 250 and call for studies on PGF metabolism, but this would be a good spot to also caution readers that the reported release rates/ratios may not be ecologically relevant.
-We agree. A sentence was added to the revised Discussion, suggesting these data must be interpreted with caution. Thanks for this suggestion.